# An Improved Altimeter in-Orbit Range Noise-Level Estimation Approach Based on Along-Track Differential Method

**Xiaonan Liu** [1,2]📷, **Weiya Kong** [1], **Hanwei Sun** [1,*], **Yongsheng Xu** [3] and **Yaobing Lu** [1]

1   Beijing Institute of Radio Measurement, Beijing 100854, China
2   The Graduate School of Second Academy of China Aerospace, Beijing 100854, China
3   Laboratory for Ocean and Climate Dynamics, Institute of Oceanology, Chinese Academy of Sciences, Qingdao 266071, China
*   Correspondence: sunhw12@tsinghua.org.cn; Tel.: +86-13488682636

**Abstract:** Satellite radar altimeters are advanced remote sensing devices that play an important role in observing the global marine environment. Accurately estimating the noise level of altimeter in-orbit ranging data is crucial for evaluating the payload performance, analyzing sea conditions, and monitoring data quality. In this study, we propose an approach based on the differential processing of along-track odd–even data sequences for altimeter in-orbit range noise-level estimation. Using the long-term along-track data sequence can notably improve the issue in the existing method in that the noise level is underestimated owing to the utilization of a relatively short data segment. On the basis of an analysis of the influence of low-frequency components on noise-level estimation, the mathematical formulas of the above differential method were deduced, and the efficacy of the approach in assessing the noise level of altimeter in-orbit data was demonstrated by simulation experiments. This method was used to estimate the noise levels of the 20 Hz datasets of Jason-3 and Sentinel-6, and the idea of the time-domain difference was extended to the frequency domain. The statistical results showed that the 20 Hz noise levels at the significant wave height (SWH) = 2 m were 7.41 cm (Jason-3 low-resolution (LR) mode), 6.66 cm (Sentinel-6 LR mode), and 3.13 cm (Sentinel-6 high-resolution (HR) mode). The power spectrum density analysis further verified its accuracy. By reprocessing the 20 Hz data of Sentinel-6 into 10, 5, and 1 Hz, the effectiveness of the along-track odd–even differential method to directly evaluate the noise level of 1 Hz data was explored, and the impact of ocean signals such as swells on noise-level estimation in synthetic aperture mode was discussed.

**Keywords:** altimeter in-orbit noise level; along-track differential method; power spectrum density analysis; Sentinel-6; ocean swells

## 1. Introduction

As an important form covering more than 70% of the Earth's surface, the ocean is crucial for regulating the Earth's climate. Multiple means are employed to monitor the Earth and to study the ocean more broadly and meticulously. As advanced remote sensing devices, satellite radar altimeters can continuously observe the ocean on a global scale. By retracking the radar echo waveform, vital parameters such as sea-level anomaly (SLA), significant wave height (SWH), and backscatter coefficient can be retrieved from the echo delay, slope, and amplitude associated with other information [1]. These parameters can be applied to research on ocean dynamics, atmospheric environment, sea ice detection, etc., after further geophysical and transmission delay corrections [2].

To utilize the above parameters successfully, the accuracy of the altimeter measurements must be guaranteed. However, there is a long data-processing link from the altimeter raw echo to the final measurement result, and various errors will be mixed, affecting the final data accuracy. These errors can be roughly divided into two categories: retracking errors

and errors of environmental transmission corrections, which mainly include tropospheric and ionospheric delays, multiple tidal and dynamic atmospheric corrections, and sea state bias [3]. Currently, mature algorithms are available for the environmental transmission correction. Most of these are empirical models based on altimeter long-term monitoring data, which have been widely verified and applied [4]. Although the magnitude (only a few centimeters) of the retracking error is very small compared to the environmental transmission disturbances, as it is located at the forefront of the data processing chain, it is the basis for all subsequent corrections. Therefore, the retracking error is a key to ensure data accuracy.

Retracking uses a mathematical algorithm to fit an echo model to the actual waveform and achieve the most consistent state, thus retrieving parameters contained in the echo. In practical applications, the sea surface height (SSH), SWH, and backscattering coefficient are obtained by retracking the altimeter waveform, which can be further studied by oceanographers [5]. So far, the least-squares algorithm has been widely adopted owing to its robustness and high accuracy [6]. According to statistical principles, the theoretical precision of waveform retracking can be calculated [7]. As sea surface speckle noise is the determinant factor of retracking precision [8], estimating the noise level of altimeter data is an effective measure for assessing the quality of retracking. Moreover, an accurate estimation of the noise level is also crucial for evaluating the payload performance, analyzing sea conditions, and monitoring data quality.

In the early stages of altimeter development, researchers usually processed data series pairs of repeated satellite orbit observations, performed power spectrum density (PSD) analysis on their differential results, and estimated the noise level on the basis of the high-frequency part [9,10]. The purpose of performing differential processing on repeated orbit observations was to remove low-frequency correlations in the data, such as the geoid, mean sea surface (MSS), large-scale ocean signals, and environmental transmission disturbances. This method has been successfully applied to several altimeter satellites, such as SeaSat, GeoSat, ERS-1, and Topex/Poseidon, providing valuable results for early noise-level estimation [11–13].

However, repeated orbit observations are at least a few days apart even in the satellite early calibration stage, which leads to the fact that, although the geographic location of the altimeter observations is the same, sea conditions change considerably. Therefore, the temporal correlation of the low-frequency signals in the data is poor, and there could still be significant residual signals in the data after repeated orbit differencing, delivering less accurate estimation of the noise level. When the satellite is in operation, the orbit repeat cycle can range from several weeks to several months, which greatly reduces the quantity of repeated orbit observation data which meet the requirements; thus, it is not conducive to long-term monitoring of the altimeter noise level. Therefore, this method has rarely been used since the 21st century. Instead, the altimeter noise level is directly estimated using the along-track data through frequency- and time-domain methods.

The frequency-domain method analyzes the power spectrum of the altimeter along-track data and estimates the noise level on the basis of the high-frequency part. This method does not remove low-frequency components via differential processing because spectral analysis can naturally separate signal components of different frequencies; thus, the noise-level estimation result can be obtained if an appropriate high-frequency noise part is selected [14,15]. Using this method, Fu et al. [16] evaluated that the 1 Hz noise levels at SWH = 2 m were 1.7 cm for Topex and 2 cm for Poseidon. Zanifé et al. [17] considered that the 1 Hz noise levels of Topex, Poseidon, and Poseidon-2 were equal to 1.6 cm after reprocessing the data using the same algorithm. Tran et al. [18] estimated that the noise level of Poseidon-3B was 1.40 cm. It should be noted that the 20 Hz data were used in the above PSD analysis, and the calculated noise level was divided by $\sqrt{20}$ for 1 Hz result. This processing assumes that the noises between the 20 Hz measurement samples are uncorrelated [19–21]. The noise level will be affected by "hump" if performing PSD analysis directly on 1 Hz data, and the estimation result will not reflect the true noise

level [22]. Moreover, owing to the high requirements for the spatiotemporal continuity of data in spectral analysis, the amount of eligible data is not large.

The time-domain method usually uses the altimeter data at 20 Hz, takes 20 samples within 1 s for analysis, performs linear fitting on it to remove the low-frequency components, and then calculates the standard deviation of the residual data as the noise level [23]. The 1 Hz accuracy typically considered for altimeters can be obtained by dividing the above result by $\sqrt{20}$. This method is simple to operate, has low data requirements, and can quickly process large amounts of altimeter data to obtain evaluation results. Using this method, Tran et al. [24] evaluated that the noise levels at SWH = 2 m were 1.8 cm for Topex and 2 cm for Poseidon. Garcia et al. [2] estimated that the altimeter noise levels of GeoSat, ERS-1, and EnviSat were 1.97 cm, 2.09 cm, and 1.76 cm, respectively, the noise level of Poseidon-2 was 1.70 cm, and the noise levels of the SIRAL altimeter on CryoSat-2 were 1.45 cm in low-resolution mode (LRM) and 1.11 cm in synthetic aperture radar (SAR) mode. Calafat et al. [25] assessed that, in pseudo-LRM mode, the noise level of SIRAL was 2.28 cm. Jiang et al. [26] deemed that the altimeter noise level of the HY-2B satellite was 1.38 cm, which was lower than that of Poseidon-3 and Poseidon-3B with 1.6 cm.

However, according to statistical principles, too few samples cannot reflect the real data characteristics. Linear fitting of only 20 sample points can easily lead to overfitting, resulting in a lower estimated noise level. If linear fitting is used in a long-term along-track series, the low-frequency components in the data cannot be completely removed, because the longer the time series, the more likely it is to introduce low-frequency components above the second order, such as changes in the geoid and variations in the sea surface itself caused by ocean dynamic phenomena [27]. This will cause the residual data to still contain some low-frequency components, and the statistical standard deviation will be higher than the true noise level. To solve the above problems, this paper proposes an altimeter in-orbit range noise level estimation approach based on the differential processing of along-track odd–even data sequences. Simulation experiments and satellite data processing were used to demonstrate the effectiveness of this method.

In Section 2, the error sources of altimeter SLA measurement and the influence of low-frequency components are analyzed. The improved noise-level estimation approach is proposed with mathematical formulas derived. The advantage of this method is verified by Monte Carlo simulation experiments, and the satellite datasets and edit criteria used in this study are elaborated. In Section 3, the noise levels of Jason-3 SSH data and Sentinel-6 SLA data are estimated, demonstrating the superiority of the along-track differential method in contrast to the existing methods. The idea of along-track difference is extended to the frequency domain, and the accuracy of this approach is further verified by PSD analysis. To explore the efficacy of the improved method in evaluating the noise level of 1 Hz data, the 20 Hz SLA data are reprocessed into 10, 5, and 1 Hz. By comparing the estimated noise levels with the theoretical values and analyzing the power spectrum, we study the effects of ocean swells, internal waves, and internal tides on the noise-level estimation and conclude that the SAR mode is more susceptible to the abovementioned ocean signals than the conventional mode.

## 2. Materials and Methods

### 2.1. The Principle of Along-Track Differential Method to Estimate Noise Level

One of the most important altimeter measurements in oceanography is sea level anomaly (SLA), which is the distance from the instantaneous sea surface height (SSH) observed by the altimeter to the mean sea surface (MSS). The inversion formula of SLA can be expressed as follows [28]:

$$\text{SLA} = \text{H} - \text{R} - \text{MSS} - \Delta\text{R}_{\text{trans}}, \tag{1}$$

where H is the satellite orbit height, R is the altimeter ranging result, MSS is the mean sea surface, and $\Delta\text{R}_{\text{trans}}$ denotes the various environmental corrections introduced during radar pulse transmission, including dry and wet troposphere delays, ionosphere delay, sea state

bias, ocean, solid earth, and polar tides, inverse atmospheric pressure, and high-frequency sea surface wind [29].

The altimeter ranging result is obtained by waveform retracking. Because sea surface speckle noise is the determinant factor affecting retracking accuracy, the retracking error manifests as a high-frequency noise characteristic, i.e., the noise level that needs to be estimated. However, as evident in Equation (1), the SLA not only contains retracking errors, but also mixes orbital, MSS, and multiple transmission path errors. Therefore, to accurately evaluate the altimeter in-orbit range noise level, the characteristics of other errors must be analyzed to assess their impact on the noise-level estimation.

The satellite orbit height is determined by precise orbit determination, which is a low-frequency parameter that changes slowly, and MSS represents the mean sea surface height after excluding periodic sea surface height changes over a long period. At present, the spatial resolution of MSS data is not higher than 1 min; thus, MSS is also a low-frequency component compared with the altimeter 20 Hz sampled data (the along-track sampling distance is approximately 300 m). However, the orbit height can change by more than 10 km in one period, and the global variation range of the MSS is approximately −100 to 100 m, with a changing rate that can reach several centimeters per kilometer. Therefore, the magnitudes of variation in both orbit height and MSS have a considerable impact on the noise-level estimation [30].

For the aforementioned multiple environmental disturbances, their own variation frequencies are relatively low, as are the spatiotemporal resolutions of the existing correction algorithms. For example, the highest spatial resolution of the global tide model is only 2 min [31]; the spatial and time resolutions for the French AVISO global grid data (for inverse atmospheric pressure correction) and the European Center for Medium-Range Weather Forecast (ECMWF) model data (for dry troposphere delay correction) are both 15 min × 15 min and 6 h, respectively [32]. The sub-satellite point spacing of a space-borne microwave radiometer (for wet troposphere delay correction) is usually 20–30 km [33]. However, the spatially varying scale and magnitude of the low-frequency disturbances have a considerable influence on noise-level estimation. For instance, the sea level change caused by ocean tides can reach several meters; the magnitude of tropospheric delay is approximately 2 m, where the dry and wet troposphere components account for about 90% and 10%, respectively; the dynamic range of inverse atmospheric pressure can exceed 10 cm, and the solid earth tide can change by 2 cm [34].

In summary, when estimating the noise level of altimeter in-orbit range data, a certain measure must be taken to remove the influence of low-frequency components. Considering that the change in low-frequency components is very slow, the existing methods usually uses 1 s as the analysis window, first performs linear fitting on 20 samples in the window, and then calculates the standard deviation of the residual data as the noise level. However, as linear fitting follows the minimum root-mean-square error criterion, too few samples cannot reflect the real statistical distribution of the data. Fitting on this basis leads to overfitting of the data, resulting in an underestimated noise level. Therefore, it can be inferred that a long-term data series should be used for noise-level evaluation. Considering that the along-track sampling distance corresponding to the altimeter 20 Hz samples is only approximately 300 m, the low-frequency components contained in the adjacent data on this spatiotemporal scale can be assumed to be basically unchanged. Hence, most of the low-frequency components can be removed by performing differential processing on altimeter along-track data, and the noise level can be estimated afterward.

Let the altimeter along-track data sequence be $X_k = \{x_k\}, k = 1, 2, \ldots, N$, and $X_k = L_k + H_k$, in which $L_k$ represents the correlated low-frequency components in the data, and the two adjacent points are considered to be the same; $H_k$ represents the uncorrelated high-frequency noise in the data, and its standard deviation is $\sigma$. If the along-track data are differentiated by subtracting the previous sample point from the latter sample point, the obtained differential sequence is

$$
\begin{aligned}
X_k^{d1} &= \{x_2 - x_1, x_3 - x_2, \ldots, x_k - x_{k-1}, \ldots\} \\
&= \{(l_2 - l_1) + (h_2 - h_1), (l_3 - l_2) + (h_3 - h_2), \ldots, (l_k - l_{k-1}) + (h_k - h_{k-1}), \ldots\} \\
&= \{h_2 - h_1, h_3 - h_2, \ldots, h_k - h_{k-1}, \ldots\} \\
&= H_k - H_{k-1}, k = 2, 3, \ldots, N
\end{aligned}
\tag{2}
$$

As $H_k$ and $H_{k-1}$ are two almost identical data sequences and are not independent of each other, although only the noise part remains in the final differential sequence $X_k^{d1}$, this processing method introduces correlation, resulting in a dilemma in that no valid conversion relationship can be established between the noise-level estimation result of $X_k^{d1}$ and the true noise level $\sigma$ of $X_k$. This means that, although we can calculate the noise level of $X_k^{d1}$, we do not know how to transform it to the noise level of $X_k$.

To solve this problem, we adopted a solution of performing differential processing on along-track odd and even sequences. That is, by subtracting the previous odd-numbered sample points from the even-numbered sample points in the data sequence, the differential sequence can be obtained as

$$
\begin{aligned}
X_k^{d2} &= \{x_2 - x_1, x_4 - x_3, \ldots, x_{2k} - x_{2k-1}, \ldots\} \\
&= \{(l_2 - l_1) + (h_2 - h_1), (l_4 - l_3) + (h_4 - h_3), \ldots, (l_{2k} - l_{2k-1}) + (h_{2k} - h_{2k-1}), \ldots\} \\
&= \{h_2 - h_1, h_4 - h_3, \ldots, h_{2k} - h_{2k-1}, \ldots\} \\
&= H_{2k} - H_{2k-1}, k = 1, 2, \ldots, \tfrac{N}{2} \, or \, \tfrac{N-1}{2}
\end{aligned}
\tag{3}
$$

As $H_{2k}$ and $H_{2k-1}$ are extracted from different parts of $H_k$ and there is no data crossover between the two sequences, they are two independent noise sequences, each with a standard deviation of $\sigma$. According to the independent and identical distribution theorem, the standard deviation of $X_k^{d2}$ increases by a factor of $\sqrt{2}$. A reliable method for estimating the noise level of altimeter along-track data is obtained, and the processing steps are as described below.

The 20 Hz along-track data sequence $X[k]$ of a certain time length is first divided into odd sequence $X_o[k]$ and even sequence $X_e[k]$. If the lengths of the two sequences are inconsistent, the last sample of the longer sequence is removed. The odd sequence is then subtracted from the even sequence to obtain the differential sequence $X_d[k]$:

$$
X_d[k] = X_e[k] - X_o[k].
\tag{4}
$$

To further remove possible residual low-frequency components, the differential sequence is linearly fitted to obtain the fitted sequence $X_f[k]$. Then, we subtract the fitted sequence from the differential sequence to obtain the residual sequence $X_r[k]$:

$$
X_r[k] = X_d[k] - X_f[k].
\tag{5}
$$

If the standard deviation of the residual sequence is calculated to be $\sigma_0$, owing to differential processing, the final noise level of this 20 Hz data sequence becomes

$$
\sigma = \sigma_0 / \sqrt{2}.
\tag{6}
$$

### 2.2. Monte Carlo Simulation to Verify the Along-Track Differential Method

2.2.1. Simulation Experiment 1

The analysis in Section 2.1 pointed out that the linear fitting of only 20 samples in the existing method can easily lead to overfitting, which makes the noise level underestimated. To verify the above conclusion, Monte Carlo simulation experiments were performed using white Gaussian noise (the mean value was 0). The data duration of a single experiment was 300 s, the data rate was 20 Hz, and the standard deviation was set as 5. The processing steps were as follows: first, the data were divided into equal-time segments, with the duration varying from 1 s to 150 s, with a 1 s step size. Then, linear fitting was performed on each data segment, and the standard deviation of the residual data was calculated. Finally, all

standard deviations were averaged to obtain the noise-level results of this experiment. We performed 100 Monte Carlo experiments as described above and considered the mean value of all results as the final noise level.

Figure 1a shows the simulation data within 20 s of an experiment and the linear fitting results of the segments of different durations. Notably, the fitting result of the 1 s data is obviously affected by the sample distribution of this segment, and the slopes of the fitting lines gradually approach zero as the segment becomes longer. The noise-level estimation results for different segment durations after 100 Monte Carlo experiments are shown in Figure 1b. When the segment duration is 1 s, the estimated result is 4.798, which is significantly smaller than the true value of 5. When the segment duration reaches 20 s, the estimated noise level is stable near the true value, with a mean value of 4.9964 and a standard deviation of 0.0051. This confirms that the noise-level analysis cannot be based on short-term data segments but on long-term data sequences of 20 s or more.

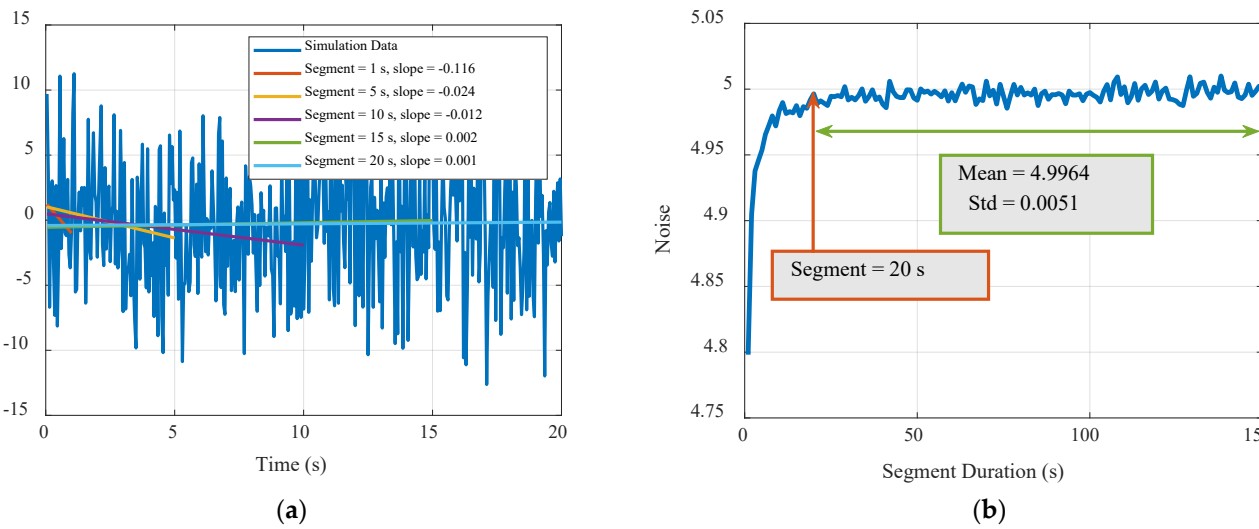

**Figure 1.** Simulation result of experiment 1: (**a**) the white Gaussian noise (blue curve) and linear fitting results (colored lines); (**b**) the estimated noise level results under different segment durations.

### 2.2.2. Simulation Experiment 2

To eliminate the influence of low-frequency signal components in long-term data on high-frequency noise-level estimation, it is proposed in Section 2.1 that along-track data can be subjected to differential processing on odd and even sequences. Simulation experiments were conducted to test whether the above method can accurately estimate the noise level of the altimeter in-orbit data. First, continuous samples of 700 s were extracted from the raw SLA data of Sentinel-6, as shown by the blue curve in Figure 2a. It can be seen from the analysis in Section 2.1 that the SLA contains a variety of low-frequency components. The SLA data were then subjected to Butterworth low-pass filtering to obtain a composite low-frequency signal, as shown by the red curve in Figure 2a. Finally, three types of Gaussian white noise (with a mean value of 0) were superimposed on the composite low-frequency signal, with standard deviations of 1, 5, and 10 cm, to obtain the final simulation data.

The differential method (DM) and nondifferential method (NDM) were used to estimate the noise level (NL) of the simulation data. The processing steps were similar to those of simulation experiment 1, except that the step of performing differential processing on odd and even sequences was added in the DM. This section selects the results at a noise level of 5 cm for presentation, and the results for 1 and 10 cm are analogous. Figure 2b shows the differential data (blue curve) and linear fitting results (red curve) of the simulated data for a segment duration of 20 s. It can be observed that differential processing indeed removes most of the low-frequency components, as the statistical characteristic of the differential data is very close to high-frequency noise. After further removal of the

residual low-frequency components via linear fitting, the noise level was estimated on the basis of the residual data.

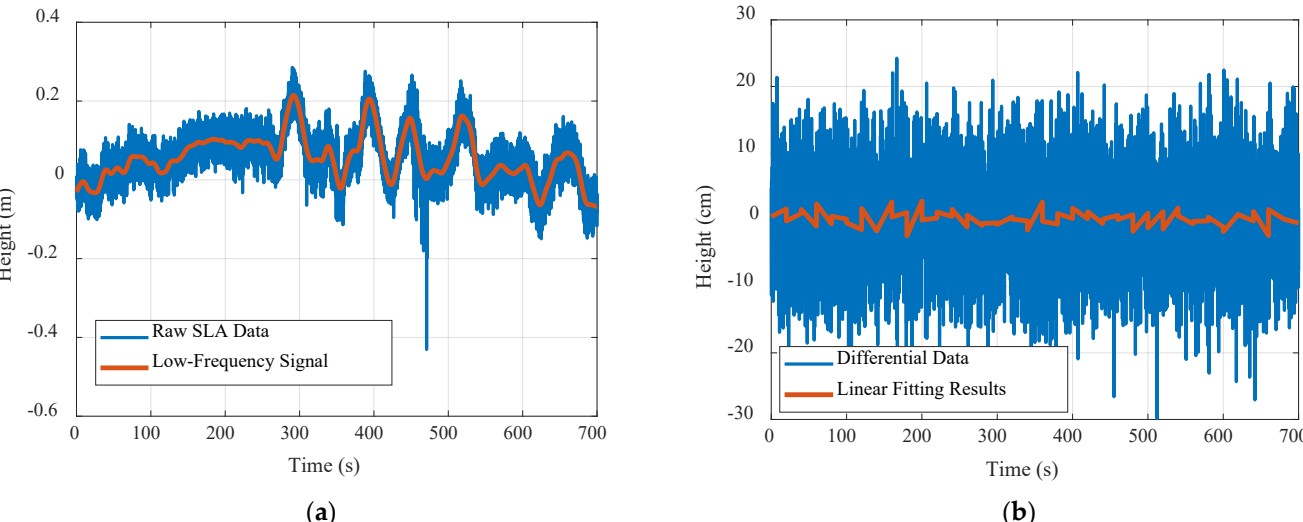

(**a**)    (**b**)

**Figure 2.** Simulation result of experiment 2: (**a**) the raw SLA extracted form Sentinel-6 (blue curve) and composite low-frequency signal obtained by Butterworth low-pass filtering (red curve); (**b**) the differential data (blue curve) and linear fitting results when the segment duration is 20 s (red curve).

The blue curve in Figure 3 shows the change in the final estimated noise level with the segment duration using the NDM, where the noise-level value corresponds to the blue vertical axis on the left, and the small window on the right is the detailed image within 6–20 s. The red curve is the estimation result obtained using the DM, and the noise-level value corresponds to the red vertical axis on the right. It can be observed that the noise level estimated by NDM increases markedly as the segment duration increases, exhibiting irregular fluctuation, and the estimation result is close to the true value of 5 cm only at approximately 12 s. The estimation result of DM is consistent with the variation shown in Figure 1b, and the NL can be accurately estimated when the segment duration approaches 20 s. This illustrates that the DM still exhibits good accuracy and robustness when the data contain a variety of low-frequency components.

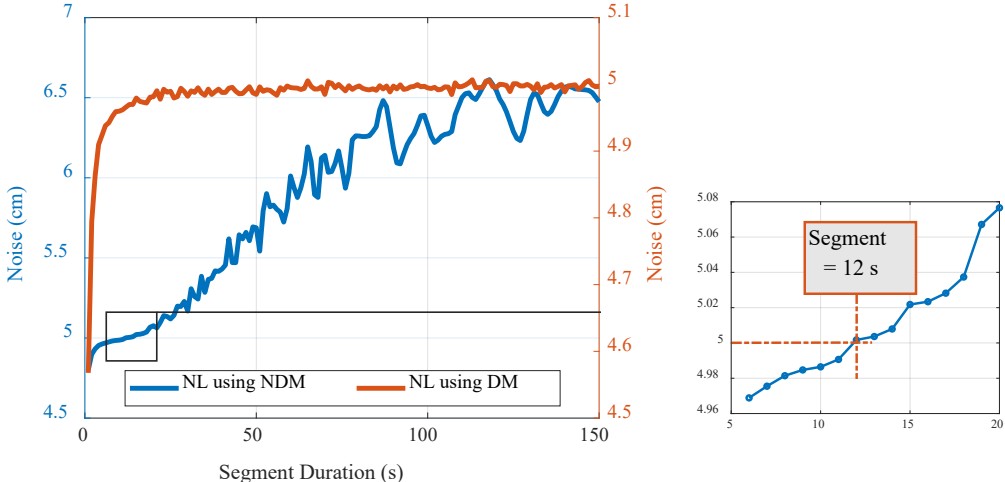

**Figure 3.** The changes in the estimated noise level (NL) with the segment duration using the non-differential method (NDM, blue curve) and differential method (DM, red curve). The small window on the right shows the detailed image of NDM within 6–20 s.

### 2.3. The Datasets and Edit Criteria of Jason-3 and Sentinel-6 Altimeters

As the successor satellite to Jason-3, Sentinel-6 has the same orbit design. The Poseidon-4 radar altimeter on Sentinel-6 adopts the "interleaved" mode (open-loop mode) for the first time, which can simultaneously acquire low-resolution (LR) and high-resolution (HR) mode data [35]. This section describes the datasets and edit criteria used for the noise-level estimation of the two satellite altimeters.

The datasets for both satellites were from Level 2 geophysical data record (GDR) products. The dataset information for the two satellites is listed in Table 1. For Jason-3, this study used the measurement data of one day in cycle 73 for analysis, with a total of 26 passes [36]. As there is no publicly released 20 Hz SLA product, the valid range data were first screened by criteria listed in Table 2, and then the corresponding orbit height data were extracted and subtracted by the range data; thus, the uncorrected raw SSH data were obtained and used for analysis. For Sentinel-6, this study used all SLA data in cycle 56 for analysis, with a total of 254 passes [37]. In addition, for Jason-3, only LR mode data were processed, while, for Sentinel-6, both LR and HR modes data were processed.

**Table 1.** The dataset information of Jason-3 and Sentinel-6 used for noise-level estimation.

| Item | Jason-3 | Sentinel-6 |
|---|---|---|
| Cycle | 073 | 056 |
| Pass | 017–042 | 001–254 |
| Mode | LR | LR and HR |
| Data | Raw SSH | SLA |
| Data rate | 20 Hz | |
| Data level | GDR | |

**Table 2.** The edit criteria of Jason-3 and Sentinel-6 GDR to ensure the data quality.

| No. | Edit Parameter | Edit Criteria |
|---|---|---|
| 5 | Surface type flag | Nonzero: $\leq$2.5% |
| 6 | SWH quality flag | Nonzero: $\leq$2.5% |
| 7 | Range quality flag | Nonzero: $\leq$2.5% |

To ensure that the data used for the analysis were not affected by anomalies or extremes, some screening conditions were employed to select valid data. Table 2 lists the seven edit criteria used in this study. Among them, No. 1 and No. 2 limit the absolute upper limits of SWH and SLA, and only the data with SWH below 10 m are accepted. No. 3 and No. 4 limit the variation ranges of two consecutive SWH and range data, which avoids the introduction of data segments containing jumps and ensures the stability of the data. No. 5 ensures that most of the data are acquired when the altimeter observation surface is open ocean. No. 6 and No. 7 further ensure the validity of the data, as flag = 0 indicates that the data quality is good.

The data processing flow for estimating the noise level of the satellite altimeter is shown in Figure 4. The basic idea is to select valid data sequences according to the "sliding window method", where the window length corresponds to the segment duration. If the data covered by the current window satisfy all the edit criteria listed in Table 2, it is determined to be a valid data sequence, the noise level is calculated using differential and nondifferential methods, and the corresponding mean SWH is recorded at the same time. Then, the window slides forward by a window length. Otherwise, it is judged to be an invalid data sequence, and the window slides forward by one sample. After reading all the data in the dataset, the statistical relationship among the estimated noise level, segment duration, and SWH is analyzed.

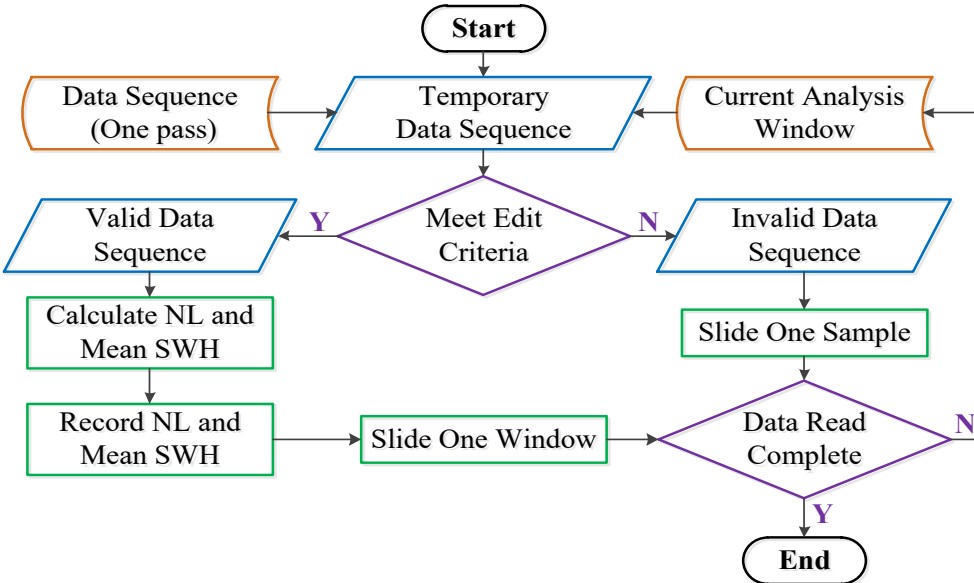

**Figure 4.** The data processing flow for estimating the noise level of the satellite altimeter.

### 3. Results

*3.1. Jason-3 Noise-Level Estimation Based on Raw SSH Measurements*

The altimeter noise level is directly related to the sea surface speckle noise, as discussed in Section 2.1. The sea state level is the main factor affecting speckle noise, and the SWH is a representative index reflecting the sea state. Studies have shown that the noise level increases with an increase in the SWH, and the two are linearly dependent [25]. In this section, the raw SSH data of the Jason-3 dataset are processed using differential and nondifferential methods, as shown in Figure 4. On the basis of verifying the above conclusion, the relationship between the noise level and segment duration was further analyzed.

Figure 5 shows two-dimensional (2D) statistical histograms of the estimated NL and SWH for both differential and nondifferential methods when the segment durations were 1 s and 30 s. In Figure 5, the horizontal axis is the SWH (m), the vertical axis is the noise level (cm), the color of the pixel unit represents the number of samples at that position, and the red curves represent the statistical median curves. When the segment duration was 1 s, the NL and SWH under the two methods were both linearly correlated, as observed from the median curves. The aggregation of the DM 2D histogram was better, whereas the NDM 2D histogram exhibited some discrete points. When the segment duration was 30 s, the statistical property of the DM was still good, and the NL and SWH showed an obvious linear correlation; however, the distribution of the NDM diverged. This demonstrates that the differential method had a good adaptability to long-term data sequences.

Figure 6a displays the statistical relationship between the NL estimated by DM and the segment duration with an SWH of 2–5 m. The variation is similar to the red curve in Figure 3 obtained from simulation experiment 2. In Figure 5, the number of samples at SWH = 2 m was the largest; therefore, the blue curve of SWH = 2 m in Figure 6a was the smoothest. All the noise level curves became stable when the segment duration was above 20 s, which matches the simulation results shown in Section 2.2. Figure 6b displays the statistical relationship between the NL estimated by NDM and the segment duration. As the segment duration increased, the noise level rose sharply; when the segment duration exceeded 30 s, the curves showed drastic fluctuation at SWH of 3–5 m.

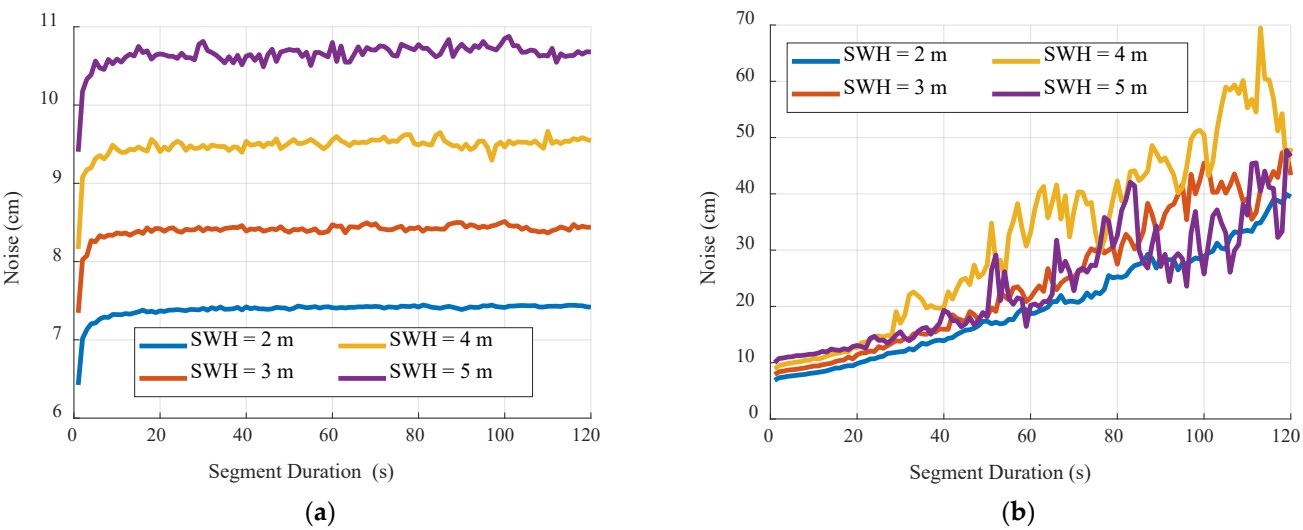

**Figure 5.** The two-dimensional statistical histograms of the estimated NL and SWH for differential method (**a**,**c**) and nondifferential method (**b**,**d**) when the segment durations were 1 s (**a**,**b**) and 30 s (**c**,**d**). The red curves represent the statistical medians.

**Figure 6.** The statistical relationships between the NL estimated using (**a**) differential and (**b**) nondifferential methods and the segment duration with an SWH of 2–5 m.

Figure 7 shows the median curves of different segment durations calculated using the differential method; notably, the noise level and SWH were linearly related. The black dashed line represents the linear fitting result of the median curve when the segment duration was 50 s, and the gray label lists the main fitting indexes. The blue curve is the noise-level estimation when the segment duration was 1 s, which was significantly lower than the final convergence curve. This again reflects that a short segment duration leads to a lower noise-level estimation, i.e., a higher ranging accuracy.

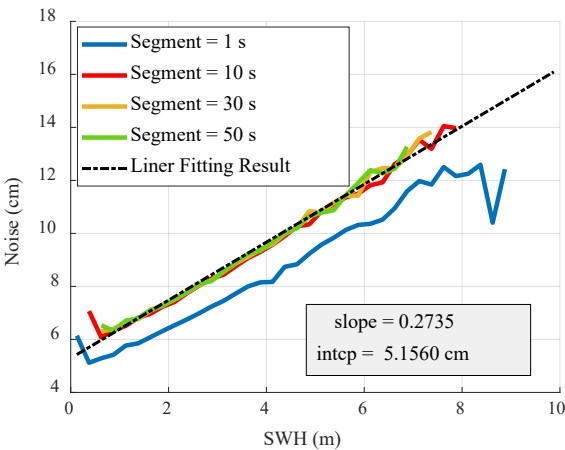

**Figure 7.** The median curves of different segment durations calculated using the differential method. The black dashed line represents the linear fitting result when the segment duration was 50 s.

### 3.2. Sentinel-6 Noise-Level Estimation Based on LR/HR SLA Measurements

In this section, SLA measurements of the LR and HR modes in the Sentinel-6 dataset are processed using the same analysis procedure as in Section 3.1. Figure 8a,b show the noise levels of the nondifferential and differential methods in the HR mode, and the results in the LR mode are shown in Figure 8c,d. In Figure 8a,c, the estimated noise levels continue to increase without stable plateaus, which are consistent with the blue curve in Figure 3. This estimation feature of the nondifferential method makes it very difficult to select an effective analysis window; therefore, the practicality of NDM for satellite in-orbit data processing is significantly decreased. Comparing Figure 8b,d with the red curve in Figure 3, it can be seen that the variation trends of the SLA noise level estimation results using the DM are very consistent with the simulation data processing result. A slight difference in Figure 8b is that the estimated NL tends to be stable when the segment duration reaches 10 s, but not 20 s. This is because the NL of the HR mode is dramatically lower than that of the LR mode; therefore, the data length required to robustly estimate the NL is shorter.

Another limitation for the selection of segment duration is that the sea state should not change significantly during the selected segment; thus, the correlation of the average SWH with the noise level is meaningful. Therefore, it is necessary to perform a statistical analysis of SWH and its changes in the dataset. Figure 9a shows the histograms of SWH in the LR and HR modes, in which the horizontal axis is the SWH (m) and the vertical axis is the probability density function (PDF). In each mode, the SWH presents a biased Gaussian distribution, with a mean value ($\mu$) of 2.657 m in the LR mode and 2.975 m in the HR mode. The standard deviations (STD) of SWH in the data segments of different durations were calculated, and the results are shown in Figure 9b. The blue and red parts represent the LR and HR mode results, respectively. The two thick curves represent the mean values of the SWH STD, and the vertical thin line bars indicate the $3\sigma$ confidence intervals of the SWH STD. Notably, the STD of SWH in the HR mode was smaller than that in the LR mode for all segment durations, which confirms that the HR mode had a higher accuracy of SWH measurement. Meanwhile, when the segment duration was in the range of 10–20 s (the black box in Figure 9b), the STD of SWH in the two modes was less than 0.5 m, which indicates that the sea state did not change

drastically in this time span. Therefore, it was reasonable to select the data sequence of this time length to estimate the altimeter noise level.

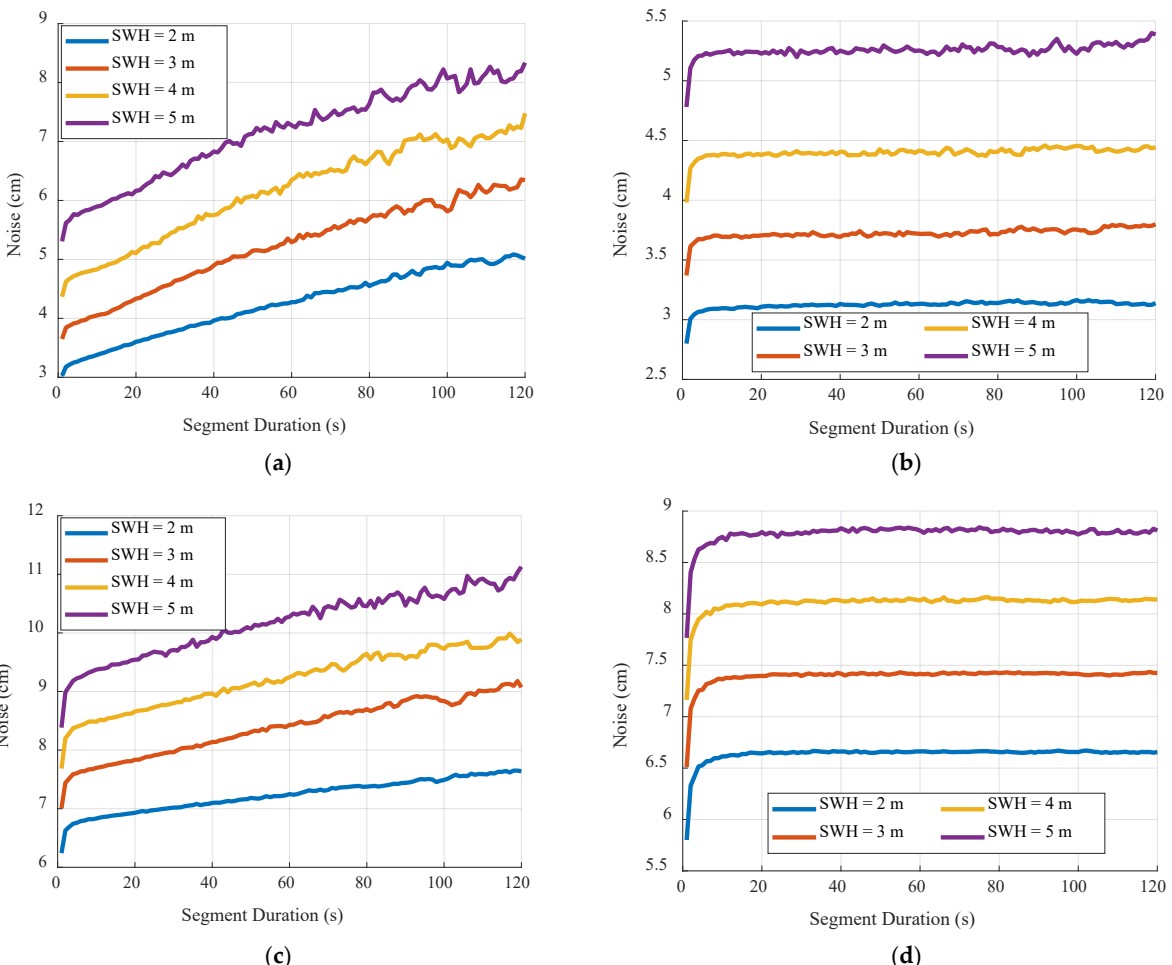

**Figure 8.** The NL of Sentinel-6 in HR (**a**,**b**) and LR (**c**,**d**) modes estimated using (**a**,**c**) nondifferential and (**b**,**d**) differential methods in relation with the segment duration with an SWH of 2–5 m.

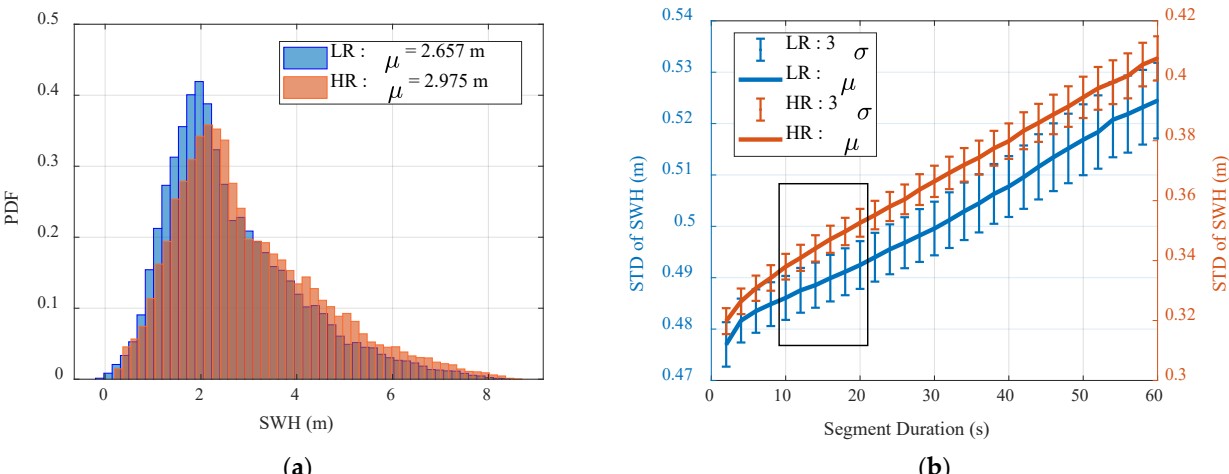

**Figure 9.** The statistical analysis results of SWH and its changes in the Sentinel-6 dataset: (**a**) the histograms of SWH in LR mode and HR mode; (**b**) the STD of SWH under different segment durations. The two thick curves stand for the mean values, and the vertical thin line bars indicate the $3\sigma$ confidence intervals.

For the SLA data of Sentinel-6 in LR and HR modes, linear fitting lines similar to those in Figure 7 were also obtained. On the basis of the statistical results, Table 3 summarizes the noise levels of Jason-3 and Sentinel-6 estimated using the differential method. In Table 3, the left three columns are the 20 Hz noise levels based on direct estimation, and the right three columns are the corresponding 1 Hz theoretical noise levels obtained by dividing the 20 Hz results by $\sqrt{20}$, assuming that the noises between the 20 Hz samples were uncorrelated.

**Table 3.** The noise levels of Jason-3 and Sentinel-6 (S6) estimated using the differential method.

| SWH | Jason-3 | S6-LR | S6-HR | Jason-3 | S6-LR | S6-HR |
|---|---|---|---|---|---|---|
| | | 20 Hz | | | 1 Hz | |
| 1 m | 6.56 cm | 5.85 cm | 2.47 cm | 1.47 cm | 1.31 cm | 0.55 cm |
| 2 m | 7.41 cm | 6.66 cm | 3.13 cm | 1.66 cm | 1.49 cm | 0.70 cm |
| 3 m | 8.47 cm | 7.42 cm | 3.72 cm | 1.89 cm | 1.66 cm | 0.83 cm |
| 4 m | 9.56 cm | 8.13 cm | 4.42 cm | 2.14 cm | 1.82 cm | 0.99 cm |
| 5 m | 10.80 cm | 8.83 cm | 5.25 cm | 2.42 cm | 1.97 cm | 1.17 cm |
| 6 m | 11.88 cm | 9.45 cm | 6.17 cm | 2.66 cm | 2.11 cm | 1.38 cm |

The 1 Hz results in Table 3 show that the noise level of Jason-3 estimated in this study was 1.66 cm at SWH = 2 m, which is slightly higher than the estimates of Jason-1, Jason-2, and Jason-3 in [2,26]. This is because the abovementioned references were all based on statistical analysis of a 1 s segment duration, yet, as we demonstrated here, the results based on 1 s are lower than the real noise level. The estimation results of the Sentinel-6 LR mode indicate that, under the same sea conditions, the noise levels of Sentinel-6 were lower than those of Jason-3, which confirms that Sentinel-6 achieved better altimeter performance and higher data quality. In the Sentinel-6 HR mode, the estimated noise level at SWH = 2 m was 0.70 cm, which meets the design requirement of Sentinel-6 [35] and is close to the evaluation result of [38]. This noise level is superior to all previous conventional altimeters and the SRAL altimeter on the Sentinel-3 satellite, which also operates in the SAR observation mode [38,39]. This result demonstrates that the open-loop SAR mode has a significant advantage over the closed-loop SAR mode.

### 3.3. PSD Analysis of Sentinel-6 SLA Data and Noise-Level Estimation

To further verify the accuracy of the estimation results in Table 3, the along-track differential method was extended to the frequency domain, and the power spectrum density (PSD) was used for the analysis. First, the eligible data sequences in the Sentinel-6 dataset were selected on the basis of Table 2, the corresponding power spectra were obtained using differential and nondifferential methods, and all power spectra in the same SWH range were averaged to obtain the ultimate result. Figure 10a,b show the average PSD with an SWH of 2–5 m in the LR and HR modes, respectively. The four near-horizontal curves in each figure represent the PSD of the differential method, and the other four downward-trending curves represent the PSD of the nondifferential method.

To ensure that the data sequence for spectral analysis had sufficient sampling points and that the sea condition did not change significantly, the length of the segment selected in this section was 60 s. It can be seen from Figure 9b that the STD of the SWH under this segment duration does not exceed 0.55 m. The advantage of power spectrum analysis is that, by transforming the time sequence into the frequency domain, the signal components of different frequencies in the data can be naturally separated. The noise level was calculated by selecting the high-frequency part of the PSD as the analysis window. However, owing to the high requirements for the spatiotemporal continuity of data in spectral analysis, there were fewer eligible data segments.

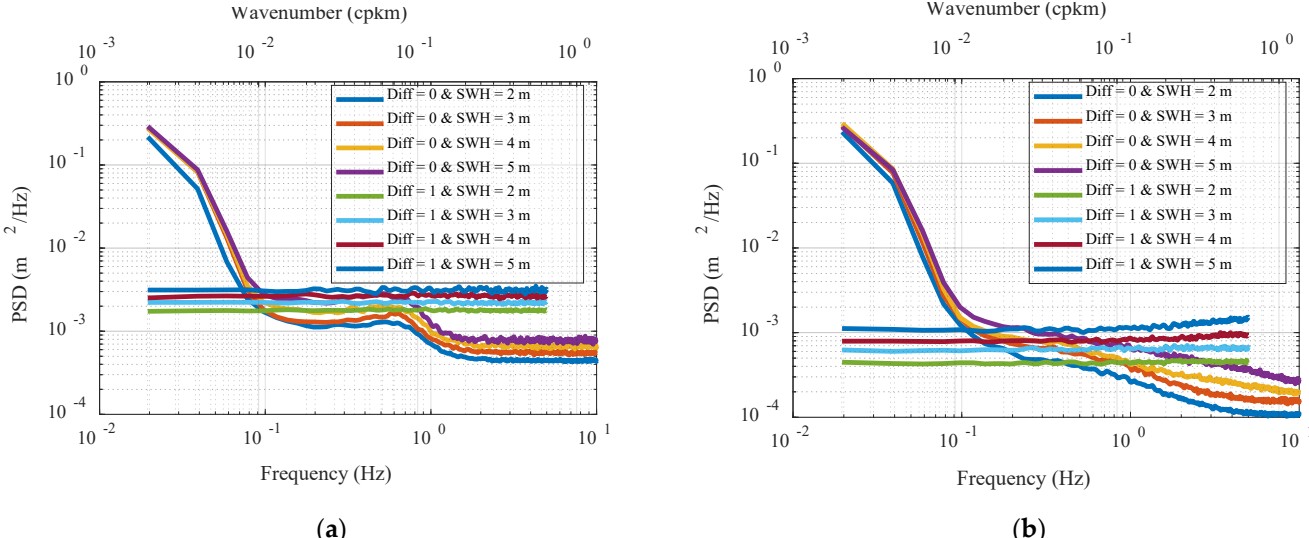

**Figure 10.** The average PSD with the SWH of 2–5 m in (**a**) LR mode and (**b**) HR mode. In each figure, Diff = 0 represents the PSD result of the nondifferential method, and Diff = 1 represents the PSD result of the differential method. The vertical axis is the power spectrum density, the bottom horizontal axis is the signal frequency, and the top horizontal axis is the wavenumber.

In Figure 10, the vertical axis is the power spectrum density, the bottom horizontal axis is the signal frequency, and the top horizontal axis is the wavenumber. It can be seen that the differential processing reduced the highest frequency by half. Owing to the characteristics of white noise, the PSD of high-frequency noise tended to remain constant. In each LR mode result of the nondifferential method, the PSD first gradually decreased to reach the first plateau, then continued to decrease, and finally reached the second plateau. These PSD curves are consistent with the previous analysis results [22,25,26]. The first plateau is caused by the nonuniformity of sea surface backscattering, which is a phenomenon that often occurs in the observations of conventional altimeters, the so-called "hump". Owing to the high-resolution characteristic of SAR altimeter, this phenomenon does not appear in the 20 Hz spectrum of HR mode [22]. In both LR and HR modes, the PSDs of the differential method are constant, which proves that most of the relevant low-frequency components are removed. As differential processing increases the noise floor by a factor of $\sqrt{2}$, in Figure 10, the high-frequency portion of the differential PSD is higher than that of the nondifferential PSD.

The noise-level estimation method based on the average power spectra was applied as follows [17]: first, we calculated the average noise power $P_n$ of the selected high-frequency interval:

$$P_n = \frac{1}{M}\sum_{f_c}^{f_B} P(f),\tag{7}$$

where $P(f)$ is the average PSD, $f_c$ is the lower cutoff frequency of the selected interval, $f_B$ is the upper cutoff frequency of the selected interval (which is equal to one-half of the sequence sampling rate $F_s$), and $M$ is the number of frequency points in the interval $[f_c, f_B]$.

The noise power of the data within the signal bandwidth was then calculated, and the noise level estimated using the differential method was obtained by

$$\sigma = \frac{1}{\sqrt{2}} \cdot \sqrt{\frac{P_n \cdot F_s}{2}}.\tag{8}$$

The PSD analysis of valid data segments in the Sentinel-6 dataset with different SWH was carried out according to Equations (7) and (8), and the noise-level estimation results were in good agreement with the corresponding values in Table 3, as shown in Table 4. This indicates that the noise-level estimation approach based on the along-track differential

method is not only suitable for the time domain but also applicable to the frequency domain. The cross-validation results in the two dimensions further demonstrated the accuracy of this improved method.

**Table 4.** The 20 Hz noise levels of Sentinel-6 (S6) estimated by time- and frequency-domain methods.

| SWH | S6-LR | S6-HR | S6-LR | S6-HR |
|---|---|---|---|---|
| | Time-Domain Method | | Frequency-Domain Method | |
| 1 m | 5.85 cm | 2.47 cm | 5.84 cm | 2.48 cm |
| 2 m | 6.66 cm | 3.13 cm | 6.65 cm | 3.12 cm |
| 3 m | 7.42 cm | 3.72 cm | 7.43 cm | 3.72 cm |
| 4 m | 8.13 cm | 4.42 cm | 8.12 cm | 4.48 cm |
| 5 m | 8.83 cm | 5.25 cm | 8.77 cm | 5.14 cm |

## 4. Discussion

In the standard product of satellite altimeters, two parallel measurement results are typically provided with the data rates of 20 Hz and 1 Hz. The previous analyses were all based on 20 Hz data; however, the 1 Hz data play an equally important role in the practical application of the altimeter. This is because other facilities that work in conjunction with altimeters, such as scatterometers and radiometers, usually only provide measurement results with data rates not higher than 1 Hz, and the various weather pattern data used for the altimeter correction are also not higher than 1 Hz. Therefore, the measurement accuracy of 1 Hz altimeter data is a more commonly used evaluation index. However, the existing 1 Hz evaluation results are all indirectly obtained from the 20 Hz noise level, on the basis of the assumption that the noises between the 20 Hz samples are uncorrelated. This is because the spatial span of dozens of 1 Hz data will reach hundreds of kilometers, and it is difficult to completely remove the changes of low-frequency components through linear fitting on such a large scale. Therefore, the evaluation results of the existing time-domain method cannot accurately reflect the true noise level.

From the previous analysis, it was demonstrated that the greatest advantage of performing differential processing on an along-track odd–even data sequence is that it can remove the low-frequency components in the altimeter in-orbit data. Therefore, this section explores whether this method can directly estimate the noise level of 1 Hz data effectively. As the sampling points drop too much from 20 Hz to 1 Hz, this section reprocesses the 20 Hz data into 10, 5, and 1 Hz for a step-by-step analysis based on the Sentinel-6 SLA dataset. First, the valid data sequences of 20 Hz were screened according to Table 2, then linear regression was performed on every 2, 4, and 20 samples to obtain 10, 5, and 1 Hz sequences, and finally, the noise levels were estimated using the along-track differential method. Table 5 lists the noise level estimation results as "estimated noise level" for the 10, 5, and 1 Hz data in the LR and HR modes (SWH of 2–4 m).

According to the premise that the noises between the 20 Hz samples are uncorrelated, the noise levels of the 10, 5, and 1 Hz data should reduce to $1/\sqrt{2}$, $1/2$, and $1/\sqrt{20}$ of the 20 Hz noise level, respectively, i.e., the "theoretical noise level" in Table 5. However, the comparison shows that the estimated noise levels of the reprocessed data are slightly higher than the theoretical values. The ratios of the estimated NL to the theoretical NL with an SWH of 2–4 m are listed in Table 5. All ratios of SWH in the range of 0.875–5.125 m were averaged; in the LR mode, the results of the 10, 5, and 1 Hz data were 1.005, 1.037, and 1.056, respectively, whereas, in the HR mode, the above results were 1.048, 1.118, and 1.195, respectively. This implies that, with a decrease in the data rate, the NL estimated by the differential method gradually becomes higher than the theoretical value, and the rate of increase is greater in the HR mode than in the LR mode.

**Table 5.** The estimated and theoretical noise levels of re-processing data and their ratios.

| Item | Rate | SWH | | | | | |
| --- | --- | --- | --- | --- | --- | --- | --- |
| | | 2 m | 3 m | 4 m | 2 m | 3 m | 4 m |
| | | LR | | | HR | | |
| Estimated Noise level | 10 Hz | 4.72 cm | 5.28 cm | 5.74 cm | 2.25 cm | 2.71 cm | 3.29 cm |
| | 5 Hz | 3.45 cm | 3.86 cm | 4.27 cm | 1.68 cm | 2.05 cm | 2.48 cm |
| | 1 Hz | 1.57 cm | 1.76 cm | 1.92 cm | 0.84 cm | 1.00 cm | 1.15 cm |
| Theoretical noise level | 10 Hz | 4.71 cm | 5.25 cm | 5.74 cm | 2.21 cm | 2.63 cm | 3.13 cm |
| | 5 Hz | 3.33 cm | 3.71 cm | 4.07 cm | 1.57 cm | 1.86 cm | 2.21 cm |
| | 1 Hz | 1.49 cm | 1.66 cm | 1.82 cm | 0.70 cm | 0.83 cm | 0.99 cm |
| Ratio | 10 Hz | 1.0021 | 1.0057 | 1.0000 | 1.0181 | 1.0304 | 1.0511 |
| | 5 Hz | 1.0360 | 1.0404 | 1.0491 | 1.0701 | 1.1022 | 1.1222 |
| | 1 Hz | 1.0537 | 1.0602 | 1.0549 | 1.2000 | 1.2048 | 1.1616 |

To analyze the reason for the above result, the PSD curves with a frequency range of 1–10 Hz in Figure 10 were enlarged, as shown in Figure 11. The left figure shows the LR mode, and the right figure shows the HR mode; the legend is the same as in Figure 10. First, we compared the nondifferential PSDs of the two modes. It can be seen that the PSDs in the LR mode remained essentially constant after a brief drop (black box in Figure 11a), which indicates that only a small amount of marine signal remained in this frequency range. However, the PSDs in the HR mode continued to decline, and clear high-frequency plateaus were only observed at SWH of 2 m and 3 m, indicating that the data still contained distinct marine signals in this frequency range. The signal wavelength corresponding to 1–10 Hz is approximately 7000–700 m, which may correspond to internal waves, internal tides, or swells. Among them, swells occur widely around the world; with strong energy and nonlinear variation characteristics, they can propagate for thousands of kilometers [40–42]. The wavelength of the swells can reach several kilometers, and it is concentrated at 300–400 m, which is exactly the azimuth resolution of the SAR altimeter in HR mode; thus, the 20 Hz sampling in HR mode can detect the swells.

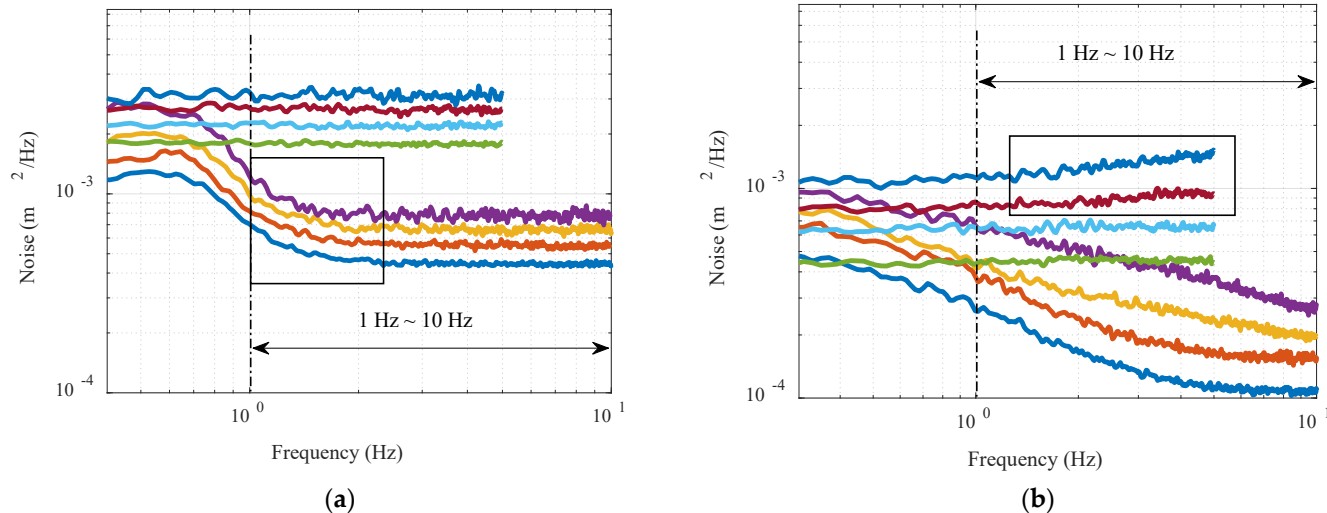

**Figure 11.** The enlarged average PSD with the frequency range of 1–10 Hz in (**a**) LR and (**b**) HR modes. The legend is the same as in Figure 10.

By reviewing the differential PSDs of the two modes, we noted that the PSDs in the LR mode were rather constant, and the PSDs in the HR mode were significantly steadier than the non-differential results, as differential processing removed most of the ocean signals. However, there were still slight lifts at the end of the power spectrum (black box in

Figure 11b), indicating that a small part of the signal components remained in the data, such as ocean swells. As the data rate was gradually decreased to 10, 5, and 1 Hz, the sampling of the signal became increasingly sparse. Although the noise level was reduced by linear regression, the variation trend of the swells could not be correctly reflected from the data and, therefore, could not be effectively removed through along-track differential processing, which eventually led to the estimated noise level gradually becoming higher than the theoretical value. The data rate of the LR mode also reached 20 Hz (the corresponding azimuth sampling interval is approximately 300 m), but its beam footprint of more than 10 km was much larger than the sampling interval; therefore, this was only equivalent to performing spatial interpolation on the marine signal in the range of 1–10 Hz, without being able to resolve the signal. On the other hand, the HR mode achieved azimuthal beam sharpening through the synthetic aperture technique; hence, it can effectively sample and resolve ocean signals, such as swells [43,44]. When the marine signal in the range of 1–10 Hz changes rapidly in space, differential processing and linear fitting cannot remove it completely; therefore, the noise-level evaluation results of the HR mode are affected to a certain extent.

## 5. Conclusions

Estimating the noise level of satellite altimeter in-orbit ranging data is important for evaluating the payload performance and monitoring data quality. The existing method utilizes a relatively short data segment (1 s), which leads to a lower noise-level estimation, i.e., a higher ranging accuracy. On the basis of an analysis of the influence of low-frequency components on noise-level evaluation, this paper proposed an altimeter in-orbit range noise level estimation approach based on the differential processing of along-track odd–even data sequences. By using long-term along-track data, the accuracy and robustness of the estimation results could be greatly improved. The main conclusions are as follows:

(1) The error sources of the altimeter in-orbit ranging data were studied, the variation scale and magnitude of the low-frequency components were thoroughly analyzed, and it was demonstrated that the low-frequency components had a significant impact on the noise-level estimation.

(2) An improved noise-level estimation approach based on the along-track differential method was proposed, and it was verified by simulation experiments that performing the differential processing on along-track odd–even sequences with long durations (approximately 20 s) could effectively remove low-frequency components in the altimeter data.

(3) Applying the above method combined with PSD analysis to estimate the noise level of Jason-3 and Sentinel-6 ranging data, the statistical results showed that the 20 Hz noise levels at SWH = 2 m were 7.41 cm (Jason-3), 6.66 cm (S6-LR mode), and 3.13 cm (S6-HR mode).

(4) By reprocessing the 20 Hz SLA data of Sentinel-6 into 10, 5, and 1 Hz, the effectiveness of the along-track odd–even differential method to directly evaluate the noise level of 1 Hz data was explored, and the ratios of estimated noise levels and theoretical noise levels were given.

(5) By analyzing the variation trend of the power spectrum in the range of 1–10 Hz, a reasonable explanation for the slightly higher noise-level evaluation result of the 1 Hz data was discussed, and it was concluded that the SAR mode was more susceptible to ocean signals such as swells than the conventional mode.

**Author Contributions:** Conceptualization, X.L. and H.S.; methodology, X.L. and W.K.; software, X.L.; validation, X.L., Y.X. and H.S.; formal analysis, X.L.; investigation, X.L. and W.K.; resources, H.S. and Y.X.; data curation, W.K.; writing—original draft preparation, X.L.; writing—review and editing, W.K.; visualization, X.L.; supervision, H.S. and Y.X.; project administration, H.S. and Y.L.; funding acquisition, H.S. and Y.X. All authors have read and agreed to the published version of the manuscript.

**Funding:** This research was funded by Science and Technology Innovation Project of LaoShan Laboratory (No.LSKJ202201301), National Natural Science Foundation of China (No.U22A20587).

**Data Availability Statement:** Not applicable.

**Acknowledgments:** The authors would like to acknowledge the support of AVISO and NASA for open access of satellite altimeter data.

**Conflicts of Interest:** The authors declare no conflict of interest.

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
