# Peer review of "An Improved Altimeter in-Orbit Range Noise-Level Estimation Approach Based on Along-Track Differential Method"

_remotesensing, doi:10.3390/rs14246250_

Round 1
Reviewer 1 Report
Good paper, well structured and with good discussions and clear conclusions. A few minor comments for improving / clarifying even further:
L76: The paragraph addresses repeated orbit observations and highlights the issues linked to the duration of the cycle and the time between observations. However, a method commonly used consists in exploiting cross-overs (comparisons on the crossing between ascending and descending orbits) within a cycle. Such comparisons are made on the same point and in time intervals shorter than the orbital cycle. Could you comment and possibly address cross-over comparisons in this section.
L151: Eq.1 does not seem correct or the contributors are not clear. Why is there a minus sign before DeltaRtrans if it is an error (i.e. residual error after correction of the effects) and not a +/- instead? Adding absolute values and errors does not seem correct. Or is this term an additional range term induced by atmospheric delays? In this case the troposphere and ionosphere "errors" should be called "delays". But again, the other contributors (e.g. tides, atmospheric pressure, etc) can be positive or negative. Please clarify the definition of the "error" terms (errors or additional delays) or correct the formula.
L168: ...the azimuth resolution is approximately 300m... I think it is meant that the along-track sampling distance is 300m. Not to be confused with the resolution which for a conventional altimeter is several km (antenna footprint) and can only be reduced down to 300m (or even less) with synthetic aperture techniques on newer generation altimeters.
L197: same as L168: confusion between resolution and sampling distance.
L209-213: Not clear what is the issue with the "final differential data". It could maybe better explained e.g. using indexes to identify what these "final" data are.
L335: Jason series are uncorrected SSH. This means that the signal contains contributions from higher frequency disturbances, like for example the ionospheric delay. If uncorrected, a fraction of this signal will remain unfiltered and will be mixed with the altimeter noise. Could this affect the conclusions and comparisons shown in table 3 (noise level based on SSH data for Jason-3 vs SLA data from S6).
Reviewer 2 Report
Please check the attachment.
